# Comprehensive Separation Algorithm for Single-Channel Signals Based on Symplectic Geometry Mode Decomposition

**DOI:** 10.3390/s24020462

**Published:** 2024-01-11

**Authors:** Xinyu Wang, Jin Zhao, Xianliang Wu

**Affiliations:** Key Lab of Intelligent Computing & Signal Processing, Ministry of Education, Anhui University, Hefei 230601, China; p22201007@stu.ahu.edu.cn (X.W.); zhaojin@ahu.edu.cn (J.Z.)

**Keywords:** FastICA, SGMD, Symplectic Geometry Component, signal separation

## Abstract

This paper aims to explore the difficulty of obtaining source signals from complex mixed signals and the issue that the FastICA algorithm cannot directly decompose the received single-channel mixed signals and distort the signal separation in low signal-to-noise environments. Thus, in this work, a comprehensive single-channel mixed signal separation algorithm was proposed based on the combination of Symplectic Geometry Mode Decomposition (SGMD) and the FastICA algorithm. First, SGMD-FastICA uses SGMD to decompose single-channel mixed signals, and then it uses the Pearson correlation coefficient to select the Symplectic Geometry Components that exhibit higher correlation coefficients with the mixed signals. Then, these components are expanded with the single-channel mixed signals into virtual multi-channel signals and input into the FastICA algorithm. The simulation results show that the SGMD algorithm could eliminate noise interference while keeping the raw time series unchanged, which is achievable through symplectic geometry similarity transformation during the decomposition of mixed signals. Comparative experiment results also show that compared with the EMD-FastICA and VMD-FastICA, the SGMD-FastICA algorithm has the best separation effect for single-channel mixed signals. The SGMD-FastICA algorithm represents an improved solution that addresses the limitations of the FastICA algorithm, enabling the direct separation of single-channel mixed signals, while also addressing the challenge of proper signal separation in noisy environments.

## 1. Introduction

The widespread deployment of communication base stations, broadcasting stations, radar equipment, and similar technologies in modern life has resulted in the creation of complex signal interference environment. The source signals normally emitted by the signal transmission equipment in such a complex signal transmission environment will be subject to a certain degree of interference. As a result, signal receiving equipment often captures not the source signal, but the unknown mixture of signals generated within the intricate signal transmission environment. To extract the desired information from the unknown mixed signals, it becomes necessary to employ signal separation technology. Thus, the scientific research goal of separating the source signal from the complex and unknown mixed signals and achieving their accurate restoration holds significant scientific importance.

The FastICA algorithm [1], which is widely used in the blind source separation of signals, has the advantages of good stability and small computation. However, this method has the drawback of being unable to directly decompose single-channel signals. Considering single-channel signal separation from the perspective of time-frequency analysis, Empirical Mode Decomposition (EMD) is an adaptive signal time-frequency analysis method [2,3]. Variational Mode Decomposition (VMD) is a signal separation algorithm that extends the classical Wiener filter to multiple adaptive frequency bands [4]. Meanwhile, the tensor analysis method and the principal component analysis method [5] address the issue of extended signal separation duration. And based on the application of EMD and VMD to signal separation, the EMD-FastICA [6] and VMD-FastICA [6,7] synthesis algorithms have been applied to solve the problem of single-channel mixed-signal separation in recent years. The EMD-FastICA algorithm initially performs EMD on the signals, which adaptively and comprehensively breaks down the signals into multiple signal components and a non-reducible trend term. Then, the FastICA algorithm is employed to decompose the multichannel signal, comprising both signal components and trend term components, and finally filters the separated signal from the signal components. However, the EMD-FastICA synthesis algorithm may notably introduce redundant components at other frequencies unrelated to the source signal. This can increase the computational complexity of the signal synthesis separation algorithm and may slightly impact the accuracy of the synthesis separation process due to noise interference. The VMD-FastICA synthesis algorithm initially obtains the mode component signals by using the Variational Mode Decomposition, and the separated signals can be obtained by inputting the multi-channel signals into the FastICA algorithm, but it remains unable to fully address the issue of signal distortion within a complex signal transmission environment.

The Symplectic Geometry Mode Decomposition (SGMD) algorithm, which was applied in the method proposed in the present study, was first proposed by Pan [8]. Symplectic Geometry Mode Decomposition begins by transforming the received mixed signals into the trajectory matrix and then proceeds to decompose the complex mixed signals in an adaptive manner to obtain a sequence of Symplectic Geometry Components (SGCs) through a series of symplectic geometry similarity transformations applied to the trajectory matrix. In this transformation, one-dimensional signals can be regarded as points on the symplectic manifold, which ensures that the unique structural features of the signals can be accurately captured. Additionally, the Symplectic Geometry Mode Decomposition method has the advantage of eliminating noise interference while keeping the raw time series of the signal unchanged, and it allows for accurate signal components to be obtained through adaptive decomposition, showcasing strong noise robustness, among other benefits. The FastICA algorithm has the advantages of accurate decomposition, a fast convergence speed, and the ability to directly separate multichannel mixed signals without requiring extensive prior knowledge about the source signals or the signal transmission environment. Thus, in the case of noise interference and the lack of most of the effective information, the use of the SGMD algorithm in conjunction with the FastICA algorithm within a comprehensive separation algorithm, for the direct and precise separation and subsequent restoration of source signals from complex and unidentified mixed signals, holds significant practical significance. This approach also introduces a novel perspective in the field of signal processing.

This article utilizes the time-frequency analysis (TFA) method to obtain the time-frequency representations of mixed signals, source signals, and decomposed signals in order to evaluate the separation performance of the proposed comprehensive separation algorithm and obtain relevant information before signal separation. Among the various methods used for time-frequency analysis, the Wigner Ville Distribution (WVD) is commonly used. WVD provides high time-resolution and frequency-resolution, enabling effective representation of the signal’s time-frequency characteristics. However, it is sensitive to noise [9,10]. The Hilbert-Huang Transform (HHT) consists of two main components: Empirical Mode Decomposition (EMD) and Instantaneous Frequency Estimation. HHT can adapt well to nonlinear and non-stationary signals by decomposing signals based on their local characteristics. However, it may encounter mode mixing issues during the EMD process [11,12]. The previous time-frequency analysis (TFA) method usually has the disadvantage of not being able to accurately characterize the time-frequency properties of a signal due to its low time-frequency resolution. In recent years, some improved time-frequency analysis (TFA) methods have been proposed. Among them, Synchronous compression based on short-time Fourier transform (SST) is a time-frequency analysis method with high time-frequency resolution [13]; Daubechies [13,14] combined synchronous compression technology with continuous wavelet transform and proposed the SWT method. The Synchronized Extraction Transform (SET) is a time-frequency analysis (TFA) method that retains only the time-frequency information in the STFT results that is most relevant to the time-varying characteristics of the signal [15]. Compared to previous time-frequency analysis (TFA) methods, these improved time-frequency analysis (TFA) methods have better time-frequency resolution. In this study, the General Linear Chirplet Transform (GLCT) is employed, which can generate the time-frequency representation with a more satisfactory energy concentration and higher time-frequency resolution. In addition, it has good noise robustness in signal processing [16].

This paper is organized as follows. The principle of the comprehensive separation algorithm based on SGMD is presented in Section 2. In Section 3, a performance analysis and comparison of SGMD-FastICA with the EMD-FastICA and VMD-FastICA comprehensive separation algorithms are presented; these were conducted in both noiseless and Gaussian white noise environments. An improved comprehensive separation algorithm based on SGMD is given in Section 4. Lastly, the conclusions are drawn in Section 5.

## 2. Principle of Comprehensive Separation Algorithm

### 2.1. Symplectic Geometry

Symplectic Geometry, also called Symplectic Topology, is a branch of differential geometry. It focuses on the examination of Symplectic manifolds, which are smooth manifolds endowed with a closed and non-degenerate two-form ω. Symplectic Geometry originates from the Hamiltonian formulation of classical mechanics, in which the phase space of a particular classical system has the structure of a Symplectic manifold [17]. The persistent nature of the Symplectic structure can be relied upon to ensure that it remains invariant during signal processing, thereby guaranteeing stability in the process of signal separation.

### 2.2. Symplectic Geometry Mode Decomposition

Assuming that the mixed signal to be decomposed is received by the signal receiving equipment as $x=x_{1},x_{2},\cdots ,x_{n}$, according to Taken’s embedding theorem, the trajectory matrix can be constructed by performing the time series topology equivalence method on a one-dimensional time series. Hence, the mixed signal x of a one-dimensional time series can be constructed into the trajectory matrix X [18].
(1)$$X=\left[\begin{array}{cccc} x_{1} & x_{1+\tau } & \cdots & x_{1+(d-1)\tau }\\ \vdots & \vdots & & \vdots \\ x_{m} & x_{m+\tau } & \cdots & x_{m+(d-1)\tau } \end{array}\right]$$
where d is the embedding dimension; τ is the delay time. In the present study, the embedding dimension is adaptively determined by applying the power spectral density [19], effectively eliminating the need to manually set an embedding dimension, which could potentially impact the accuracy of SGMD.

After obtaining the trajectory matrix X, autocorrelation analysis was performed on the matrix X to obtain the covariance matrix A:(2)$$A=X^{T}X$$

Following the symplectic geometry similarity transformation, the matrix $A^{2}$ can be decomposed to obtain $Q_{i}$, where $Q_{i}\left(i=1,2,\cdots ,d\right)$ is the eigenvector of the corresponding eigenvalue of the matrix $A^{2}$.

The coefficient matrix W was constructed using the eigenvector matrix Q and the trajectory matrix X:(3)$$W_{i}=Q_{i}^{T}X$$

The single component matrix Z was obtained using the eigenvector matrix Q and the coefficient matrix W:(4)$$Z_{i}=Q_{i}W_{i}$$

The single component matrix Z is transformed to obtain the symplectic geometric single-component reconstruction matrix [20]. Its conversion expression is as follows:(5)$$y_{k}=\left\{\begin{array}{ll} \frac{1}{k}\sum_{p=1}^{k}z_{p,k-p+1}^{*} & 1\leq k\leq d^{*}\\ \frac{1}{d^{*}}\sum_{p=1}^{d^{*}}z_{p,k-p+1}^{*} & d^{*}\leq k\leq m^{*}\\ \frac{1}{n-k+1}\sum_{p=k-m^{*}+1}^{n-m^{*}+1}z_{p,k-p+1}^{*} & m^{*}<k\leq n \end{array}\right.$$

Diagonal averaging of each matrix yielded a one-dimensional time series, which resulted in d set of initial single-component signals, i.e., $Y_{1},Y_{2},\cdots ,Y_{d}$.

Finally, each initial single component was reorganized, so as to remove components such as the same characteristics that existed between the groups of components. The first initial single component $Y_{1}$ was selected and its period was compared with the periods of the remaining d−1 initial single components. The first Symplectic Geometry Component ${\mathrm{SGC}}_{1}$ could be obtained by performing the initial single-component reorganization within the error tolerance. Then, the constituent initial single component signals of ${\mathrm{SGC}}_{1}$ were removed and the remaining initial single components were summed up and denoted as $g^{1}$. Finally, the first component of the remaining initial single components was selected to be analyzed with the other components for cycle similarity analysis, which could be obtained as ${\mathrm{SGC}}_{2}$. After h iterations, the h Symplectic Geometry Components could be obtained, and the summation of the remaining initial single components could be written as $g^{h}$. Its termination condition could be set as:(6)$$NMSE^{h}=\frac{\sum_{e=1}^{n}g^{h}(e)}{\sum_{e=1}^{n}x(e)}$$

As a result, the Normalized Mean Squared Error (NMSE) ratio of the residual signal could be derived, and the decomposition terminated when the obtained $NMSE^{h}$ was less than a set threshold; if not, the iterative decomposition continued. Available at the end of decomposition:(7)$$x(n)=\sum_{h=1}^{N}SGC^{h}(n)+g^{\left(N+1\right)}(n)$$

The flowchart of the Symplectic Geometric Mode Decomposition method as shown in Figure 1.

### 2.3. FastICA

The Independent Component Analysis (ICA) [21] technique is a significant method in blind source separation. ICA is an algorithm-based approach used to identify inherently independent factors within statistical data. In the present study, we use the independent component analysis algorithm, FastICA, which is based on the fast calculation of negative entropy [22,23].

First, there must be certain constraints to ensure the feasibility of the FastICA algorithmic model:Independent components are statistically independent;Independent components have non-Gaussian distributions;The number of independent source signals is equal to the number of received mixed signals.

In the independent component algorithm for fast computation based on negative entropy, the negative entropy expression for a random variable is:(8)$$J\left(x\right)=H\left(x_{gauss}\right)-H\left(x\right)$$

If only a non-quadratic function G is used, the corresponding approximation becomes:(9)$$J\left(x\right)\infty {\left\{E\left[G\left(x\right)-E\left[G\left(x_{g}\right)\right]\right]\right\}}^{2}$$

Using Newton’s iterative method, the following iterative equation can be obtained:(10)$$W_{k+1}=E\left\{xg\left(W_{k}^{T}x\right)\right\}-E\left\{g^{\prime }\left(W_{k}^{T}x\right)\right\}W_{k}$$

Through normalization, the following can be obtained:(11)$$W_{k+1}=W_{k+1}/\left\|W_{k+1}\right\|$$

Therefore, the decomposition process of FastICA algorithm for the input signal is as follows:Preprocessing: Centering and whitening.Randomly selecting and initializing $W_{p}$; $W_{p}=W_{p}/\left\|W_{p}\right\|$.Updating $W_{p}$; $W_{p+1}=E\left\{xg\left(W_{p}^{T}x\right)\right\}-E\left\{g^{\prime }\left(W_{p}^{T}x\right)\right\}W_{p}$.Normalizing $W_{p}$; $W_{p}=W_{p}/\left\|W_{p}\right\|$.Checking for convergence on the normalized $W_{p}$. If convergence does not occur, then return to step 4. If convergence occurs, then output the independent components of the algorithm decomposition.

### 2.4. Comprehensive Separation Algorithm Based on Symplectic Geometric Mode Decomposition

To address the shortcomings of previous signal processing techniques, Symplectic Geometry Mode Decomposition was adopted for a novel signal separation algorithm within the Symplectic Geometry Analysis method. SGMD is a signal separation method rooted in the geometric properties of phase space, which allows for the preservation of the signal’s system structure. At its core, SGMD involves mapping a one-dimensional time series into a phase space to create a phase space matrix. Subsequently, it leverages the symplectic matrix similarity transform within the phase space to compute the eigenvalues of the target Hamiltonian matrix. The corresponding eigenvectors are then computed through the eigenvalues so as to reconstruct and obtain the Symplectic Geometry Component (SGC), which realizes adaptive and complete signal separation for complex mixed signals received by SGMD. The Symplectic Geometry Mode Decomposition algorithm employs symplectic geometry similarity transformation during the decomposition of mixed signals, which ensures that the time series of the source signals, which constitute the mixed signals, remain unaltered throughout the decomposition process.

Addressing the limitations of the conventional blind source separation method, particularly the widely adopted FastICA algorithm, which struggles with direct separation of single-channel mixed signals and signal distortion in noisy environments, a novel single-channel signal synthesis separation algorithm was introduced. This algorithm combines the SGMD algorithm with the FastICA algorithm, and its primary aim is to resolve the challenge of efficiently separating unknown and complex mixed signals. Through the FastICA algorithm, which is based on the fast calculation of negative entropy, the received single-channel mixed signals are first subjected to time-frequency analysis (TFA), and the number of source signals before mixing is determined according to the time-frequency diagrams corresponding to the mixed signals obtained using the General Linear Chirplet Transform.

Subsequently, by applying the power spectral density adaptively, the important parameters of embedding dimension and delay time, which determine the trajectory matrix, are determined. The received single-channel mixed signal is then transformed into a trajectory matrix, and the matrix is subjected to autocorrelation analysis to obtain the covariance matrix. After a series of symplectic geometry similarity transformations on the covariance matrix, the initial single-component matrix is obtained. Then, by performing diagonal averaging on the initial single-component matrix, multiple single-component signals are obtained. Finally, in order to eliminate possible common periodic components, common frequency components, and other various identical compositions among the multiple component signals, periodic similarity is used as an evaluation metric. After the obtained multiple single-component signals have been reconstructed, a total of n SGCs are ultimately obtained, and after n SGCs are obtained by the SGMD, the Pearson correlation coefficients between these n SGCs and the mixed signals are computed separately. The k SGCs that are below the Pearson correlation coefficient threshold 0.45 are eliminated, and the remaining n−k SGCs are selected to be expanded into the virtual multichannel hybrid signal together with the original single-channel mixed signals. The number of channels in the virtual multichannel signals is determined by the number of source signals, which is initially based on the total number of time-frequency images of the different signals clearly displayed in the time-frequency diagram.

Finally, the newly constructed virtual multichannel mixed signal is fed into the FastICA algorithm to obtain the accurate separated signal. The proposed SGMD-FastICA single-channel signal comprehensive separation algorithm possesses the FastICA algorithm’s ability to quickly separate and restore the received unknown mixed signals to the source signals with good stability, fast convergence speed, and small computation volume in cases in which the signal transmission environment or the source signals’ related information are unknown. At the same time, the proposed algorithm also accommodates the SGMD algorithm’s ability to keep the time series of the original signal unchanged during the decomposition of mixed signals, with superior noise robustness in the environment of noise interference. Hence, the proposed algorithm effectively addresses the two primary challenges encountered in traditional blind source separation methods. Specifically, it resolves the issues of the FastICA algorithm’s inability to directly separate single-channel mixed signals and its limited noise resistance. The process flow of the proposed SGMD-FastICA comprehensive separation algorithm is illustrated in Figure 2.

To verify the validity of the algorithm for signal separation of received single-channel mixed signals, the correlation coefficient is chosen to evaluate the final separation effect of the comprehensive separation algorithm on the mixed signals. The correlation coefficient evaluates the separation effect with the following formula:
(12)$$\xi_{ij}=\xi \left(s_{i},y_{j}\right)=\left|\frac{\sum_{t=1}^{M}s_{i}\left(t\right)y_{j}\left(t\right)}{\sqrt{\sum_{t=1}^{M}s_{i}^{2}\left(t\right)\sum_{t=1}^{M}y_{i}^{2}\left(t\right)}}\right|$$
where $\;s_{i}\left(t\right)$ is the ith signal in the source signal and $y_{j}\left(t\right)$ is the jth signal in the decomposed signal obtained using the SGMD-FastICA comprehensive separation algorithm.

Thus, when a single-channel mixed signal is received by the signal-receiving device, the specific steps for achieving an accurate reduction in the source signal using the SGMD-FastICA comprehensive separation algorithm are as follows:Time-frequency analysis is performed on the received mixed signals, and the number of source signals is determined based on the corresponding time-frequency diagram.The power spectral density of the mixed signal is calculated to obtain the parameters of the trajectory matrix, i.e., the embedding dimensions and thus the corresponding trajectory matrix.A series of symplectic matrix similarity transformations are conducted on the trajectory matrix to obtain the SGCs of the reconstruction.The SGCs characterized by significant correlation coefficients with the mixed signal are chosen and expanded alongside the single-channel mixed signal, resulting in a new virtual multi-channel mixed signal.The multichannel mixed signal is fed into the FastICA algorithm to obtain the final decomposed signal.The effectiveness of the comprehensive separation algorithm is verified using Equation (12).

## 3. Performance Analysis of the SGMD-FastICA Comprehensive Separation Algorithm

### 3.1. Signal Separation in Noise-Free Interference Environments

In the present study, a comprehensive separation algorithm based on the combination of the FastICA algorithm and the Symplectic Geometry Mode Decomposition is used to simulate the signal separation of multiple Linear Frequency Modulation (LFM) signals and Frequency-Modulated (FM) signals.

Two LFM signals, $s_{1}\left(t\right)$ and $\;s_{2}\left(t\right)$, and FM signal $\;s_{3}\left(t\right)$ were source signals. The bandwidth of the LFM source signal $\;s_{1}\left(t\right)$ was 1 MHz, the sampling frequency was 40 MHz, and the chirp slope was 100 GHz/s. The bandwidth of the LFM source signal $\;s_{2}\left(t\right)$ was 20 MHz, the sampling frequency was 10 MHz, and the chirp slope was 2 THz/s. The FM source signal $\;s_{3}\left(t\right)$ was sampled at 1000 Hz, the carrier frequency was 100 Hz, the modulating signal was $\;sin\left(2\pi \;*\;10\;*\;t\right)$, the modulation frequency was 10 Hz, and the modulation index was 5. The recording length of all source signals was −10 microseconds to −2.5 microseconds, the amplitude of all source signals was 1, and the number of samples was 3500. The mixed signal $x\left(t\right)$ was formed by the linear combination of $\;s_{1}\left(t\right)$, $\;s_{2}\left(t\right)$ and $\;s_{3}\left(t\right)$. Its expression is as follows:(13)$$x\left(t\right)=A*S=a_{1}s_{1}\left(t\right)+a_{2}s_{2}\left(t\right)+a_{3}s_{3}\left(t\right)$$

Furthermore, $A=\left[0.8,\;1,\;0.5\right]$, $S={\left[s_{1}\left(t\right),\;s_{2}\left(t\right),\;s_{3}\left(t\right)\right]}^{T}$.

The time-domain waveforms of the LFM source signals $s_{1}\left(t\right)$, $\;s_{2}\left(t\right)$ and FM signal $\;s_{3}\left(t\right)$ used in the experiments and the single-channel mixed signals received by the signal receiving equipment assumed through the simulation experiments are shown in Figure 3.

Firstly, the single-channel mixed signal underwent time-frequency analysis, resulting in the time-frequency diagram shown in Figure 4. It is obvious from the figure that the mixed signal is a combination of three source signals.

The single-channel mixed signal was then subjected to SGMD, and after the adaptive decomposition was completed to obtain the SGCs, five SGCs with Pearson correlation coefficients greater than 0.45 were filtered out by calculating the Pearson correlation coefficients of the SGCs and the original mixed signal, respectively, as illustrated in Figure 5.

Next, a new virtual multichannel mixed signal is formed based on the original mixed signal and the five filtered-out SGCs, and the new multichannel mixed signal consists of the original mixed signal and the SGC1, SGC2, and SGC3 signals with Pearson correlation coefficients greater than 0.5 and the SGC4 and SGC5 signals with Pearson correlation coefficients ranging from 0.45 to 0.5. The new virtual multichannel mixed signals are then fed into the FastICA algorithm. Finally, the comprehensive algorithm can obtain three decomposed signals and three trend terms. In accordance with the number of source signals determined based on the time-frequency diagram of the mixed signal, we can obtain three decomposed signals corresponding to the three source signals, as depicted in Figure 6.

In the present study, we compare the waveforms, time-frequency diagrams and correlation coefficient matrices of the source signals and the decomposed signals obtained using the comprehensive separation algorithm, so as to examine the signal separation effect of the comprehensive separation algorithm in a noise-free environment. In terms of the signal waveforms, the waveforms of the source signal and the decomposed signal obtained using the SGMD-FastICA comprehensive separation algorithm were basically the same. Therefore, it can be seen that the comprehensive separation algorithm performs well when it comes to maintaining the characteristics of the source signal. Regarding the time-frequency diagrams, the shapes of the time-frequency diagrams of the source signal and the corresponding time-frequency diagrams of the separated signals were basically the same as those of the corresponding time-frequency diagrams and the colors of the different frequencies. Thus, it can be inferred that the time-frequency characteristics of the separated signals obtained after the comprehensive separation algorithm and the signal energies at different frequencies are consistent with the initial source signals.

According to the correlation coefficient matrix, the correlation coefficient of the source signal $\;s_{1}\left(t\right)$ and its corresponding decomposed signal $y_{1}\left(t\right)$ was as high as 99.79%, while the correlation coefficient of the unrelated decomposed signal $y_{2}\left(t\right)$ was only 0.001%; the correlation coefficient of the source signal $\;s_{2}\left(t\right)$ and its corresponding decomposed signal $y_{2}\left(t\right)$ was as high as 99.19%, while the correlation coefficient of the unrelated decomposed signal $y_{1}\left(t\right)$ was only 0.30%; the correlation coefficient of the source signal $\;s_{3}\left(t\right)$ and its corresponding decomposed signal $y_{3}\left(t\right)$ was as high as 97.12%, while the correlation coefficient of the unrelated decomposed signal $y_{1}\left(t\right)$ was only 0.02%. The correlation coefficient between the decomposed signal and the corresponding source signal was close to 100%, and the correlation coefficient with the uncorrelated source signal was close to 0. An observation can be made that the proposed SGMD-FastICA comprehensive separation algorithm had a significant effect on the separation of the unknown single-channel hybrid signal.

The EMD-FastICA and VMD-FastICA algorithms were selected for signal separation comparison experiments on the same mixed signals composed of the same source signals. The EMD-FastICA algorithm began by using the EMD algorithm to adaptively decompose the received mixed signals into six component signals and one trend term, and then input the component signals and trend term into the FastICA algorithm to obtain seven component signals, as shown in Figure 7. The three decomposed signals $y_{2}\left(t\right)$, $y_{3}\left(t\right)$ and $y_{5}\left(t\right)$ which were most similar to the source signal were selected. The VMD-FastICA comprehensive separation algorithm began by using the VMD algorithm to decompose the mixed signals into three IMF components according to the number of source signals, and then input these IMF components into the FastICA algorithm to obtain the final decomposed signals, as illustrated in Figure 8. Among them, the time-frequency diagrams of the decomposed signals acquired through the SGMD-FastICA comprehensive separation algorithm, as well as the EMD-FastICA and VMD-FastICA comprehensive separation algorithms, are illustrated in Figure 9. The matrices of the correlation coefficients obtained using the three comprehensive separation algorithms are shown in Table 1.

Based on all the results above, an observation can be made that although the EMD-FastICA and VMD-FastICA comprehensive separation algorithms can separate one or two signals without distortion in a noise-free environment, the separation effect is vastly inferior to that of the SGMD-FastICA comprehensive separation algorithm. Firstly, by comparing the source signal with the decomposed signals obtained using the three comprehensive separation algorithms, it can be seen that the SGMD-FastICA comprehensive separation algorithm performs the best when it comes to maintaining the characteristics of the source signal. Secondly, it can be clearly seen from the time-frequency diagrams in Figure 9 that the time-frequency characteristics of the decomposed signals obtained using the EMD-FastICA and the VMD-FastICA comprehensive separation algorithms have been changed compared with those of the source signal, and only the SGMD-FastICA comprehensive algorithm maintains the time-frequency characteristics of the source signals during the signal separation process. Lastly, the matrix of correlation coefficients in Table 1 shows that the EMD-FastICA algorithm can only successfully separate two signals, and the correlation coefficient of the third decomposed signal and its corresponding source signal is only 57.45%, which is far lower than the standard of 80%, and then it is judged that its signal separation is distorted. Meanwhile, the VMD-FastICA comprehensive separation algorithm can only separate two signals successfully, and the correlation coefficient of the second decomposed signal and its corresponding source signal is only 63.27%, and it is judged that the algorithm is also distorted for the separation of the mixed signals. Therefore, it can be concluded that in a noise-free environment, facing the reception of unknown single-channel mixed signals, the separation effect of the SGMD-FastICA algorithm is the best and the most convenient, compared with that of the EMD-FastICA and VMD-FastICA algorithms.

### 3.2. Signal Separation in Gaussian White Noise Environment

In the present experiments, Gaussian white noise served as the noise interference signal to facilitate a comparative analysis of the decomposition performance between the SGMD-FastICA comprehensive separation algorithm and the EMD-FastICA and VMD-FastICA comprehensive separation algorithms. This assessment was conducted under the condition of receiving the same mixed LFM signal-containing noise. The source signals used in the experiments were consistent with the two LFM source signals used in the aforementioned noise-free environment.

Firstly, in a noise interference environment with 20 dB SNR, the received single-channel LFM mixed signal containing Gaussian white noise is illustrated in Figure 10. Its expression is as follows:(14)$$x\left(t\right)=A*S+n(t)$$

Furthermore, $A=\left[0.8,\;1\right]$, $S={\left[s_{1}\left(t\right),\;s_{2}\left(t\right)\right]}^{T}$, n(t) is Gaussian white noise.

The decomposed signals obtained using the SGMD-FastICA comprehensive separation algorithm are illustrated in Figure 11. The decomposed signals obtained using EMD-FastICA and VMD-FastICA comprehensive separation algorithms are illustrated in Figure 12 and Figure 13, respectively. The correlation coefficients of the two decomposed signals obtained using the three comprehensive separation algorithms in a 20 dB SNR environment were calculated based on the correlation coefficient Equation (12) and the correlation coefficients calculated using the three separation algorithms are shown in Table 2. Based on the waveforms and correlation coefficients of the decomposed signals, an observation can be made that compared with the other two signal separation algorithms, the SGMD-FastICA comprehensive separation algorithm had the best effect on the separation of the noisy single-channel mixed LFM signals under 20 dB SNR, and the comprehensive separation algorithm based on the SGMD was able to eliminate the interference of noise to a certain extent.

To facilitate a more complete comparison between the SGMD-FastICA comprehensive separation algorithm and the other two signal separation algorithms in the presence of Gaussian white noise interference, three types of signal separations were conducted sequentially. These separations were conducted under varying signal-to-noise ratios, ranging from 5 dB to 20 dB. The correlation coefficients between the separated signal $y_{1}\left(t\right)$ and the source signal $\;s_{1}\left(t\right)$, and the correlation coefficients between the separated signal $y_{2}\left(t\right)$ and the source signal $\;s_{2}\left(t\right)$ are illustrated in Figure 14. An observation can be made that in the 5–20 dB SNR environment, only the correlation coefficients of both LFM signals separated using the SGMD-FastICA comprehensive separation algorithm were consistently above 80%, providing the best separation results. Although the separation effect of the first separated signal $y_{1}\left(t\right)$ separated using VMD-FastICA was similar to that of SGMD-FastICA, the correlation coefficient of the second separated signal was consistently lower than 60%. This can be attributed to the algorithm’s suboptimal performance in overall decomposition of the complex LFM mixed signal. Thus, this observation suggests that the signal separation achieved by the VMD-FastICA comprehensive separation algorithm in this scenario was distorted.

## 4. Improved Comprehensive Separation Algorithm Based on Symplectic Geometry Mode Decomposition

In an environment characterized by strong noise interference, the presence of excessive Gaussian white noise can inevitably have a detrimental impact on both the time-frequency analysis and the FastICA algorithm used within the comprehensive separation algorithm based on SGMD.

In the present study, an improved comprehensive separation algorithm based on SGMD was proposed to improve the accuracy of signal separation in a strong noise interference environment. The unknown single-channel mixed signal is first pre-processed using wavelet denoising, and the number of source signals before mixing is determined based on time-frequency analysis of the pre-processed mixed signals, which in turn determines the number of channels to be used in the subsequent construction of the virtual multi-channel signal based on the number of source signals. Subsequently, the pre-processed mixed signal is fed into the SGMD-FastICA comprehensive separation algorithm to achieve accurate signal separation in noisy environments. The process flow of the improved comprehensive separation algorithm is shown in Figure 15.

The LFM source signals used in the present experiment were consistent with the aforementioned series of experiments. Firstly, the received single-channel mixed LFM signal containing Gaussian white noise was pre-processed by means of wavelet denoising in the noise interference environment with a 0 dB signal-to-noise ratio. The noise-containing mixed signal and the pre-processed mixed signal are shown in Figure 16. The results of the time-frequency analysis of the pre-processed mixed signals show that the mixed signals were a mixture of two LFM source signals, and the number of channels of the subsequently constructed virtual multichannel signals was set to two. The time-frequency plot of the pre-processed mixed signals is illustrated in Figure 17.

The pre-processed single-channel mixed LFM signal was fed into the SGMD algorithm. The pre-processed single-channel LFM signal with the Symplectic Geometry Components was expanded into a virtual two-channel LFM signal. This newly constructed dual-channel LFM signal was then input into the FastICA algorithm, resulting in the final decomposed signal depicted in Figure 18. The time-frequency diagrams for both the decomposed signal and the source signal can be observed in Figure 19. Additionally, the correlation coefficient matrices of the decomposed signals $y_{1}\left(t\right)$, $y_{2}\left(t\right)$ and the source signals $\;s_{1}\left(t\right)$, $\;s_{2}\left(t\right)$ are shown in Table 3.

The observation that the correlation coefficient between the decomposed signal and its corresponding source signal tended to approach 100%, while the correlation coefficient with unrelated source signals tended to approximate 0, coupled with the notable similarity in the shapes of the time-frequency diagrams between the source signal and the corresponding decomposed signal, especially in terms of color patterns at different frequencies, provides compelling evidence that the improved comprehensive separation algorithm based on SGMD excels in environments characterized by strong noise interference, even with a signal-to-noise ratio as low as 0 dB.

In order to further test the performance of the enhanced comprehensive separation algorithm based on SGMD when it comes to handling single-channel mixed LFM signals under conditions of strong noise interference, the single-channel LFM mixed signals which mixed from the same LFM source signals containing Gaussian white noise were separated sequentially in a signal-to-noise ratio of −30 dB to 10 dB, the corresponding correlation coefficients between the decomposed signals $y_{1}\left(t\right)$, $y_{2}\left(t\right)$ and the source signals $\;s_{1}\left(t\right)$, $\;s_{2}\left(t\right)$ were computed using Equation (12); the correlation coefficients are shown in the line graph in Figure 20. Based on the experimental results, an observation can be made that the correlation coefficients of the improved integrated separation algorithm based on SGMD were consistently greater than 94% when comparing the separated signals with their corresponding source signals across a range of signal-to-noise ratios from −30 dB to 10 dB. This substantiates the algorithm’s ability to effectively and adaptively decompose complex single-channel LFM mixed signals and accurately recover the source signals even in challenging environments characterized by strong noise interference.

## 5. Conclusions

In the context of today’s intricate signal interference environments, it is common for signal-receiving equipment to capture not the original source signal emitted by the transmitting equipment, but rather an unknown mixed signal. In response to this challenge, a comprehensive signal-separation algorithm was proposed in the present study. This algorithm leverages a combination of SGMD and FastICA. This approach addresses the issues of the traditional FastICA algorithm in signal separation distortion in the environment of noise interference and the inability to directly separate the signals of single-channel mixed signals. In order to analyze the performance of the proposed algorithm, it was compared with the EMD-FastICA and VMD-FastICA comprehensive separation algorithms in a noise-free environment and in a Gaussian white noise interference environment, respectively. The exceptional robustness and accurate signal decomposition characteristics of the improved SGMD-FastICA integrated separation algorithm were further demonstrated in simulation experiments in a strong noise-interference environment. In brief, all the experimental results show that, in the case in which the received mixed signals have no spectrally aliased portion and a small amount of spectrally aliased portion, the SGMD-FastICA comprehensive separation algorithm proposed in this paper has the best separation effect when it comes to solving the problem of single-channel mixed signals and has the best noise robustness in noisy interference environments compared with EMD-FastICA and VMD-FastICA.

Further, the methodology for determining the embedding dimension in the construction of the trajectory matrix needs to be explored further. On the other hand, in the case of a special scenario in which the received single-channel mixed signal exhibits severe spectral aliasing, addressing the impact of severe spectral aliasing on signal separation and processing can be achieved by increasing the a priori knowledge of the signal’s sparsity or statistical properties, energy properties, and so on, which needs to be further explored in the future.

## Figures and Tables

**Figure 1 sensors-24-00462-f001:**
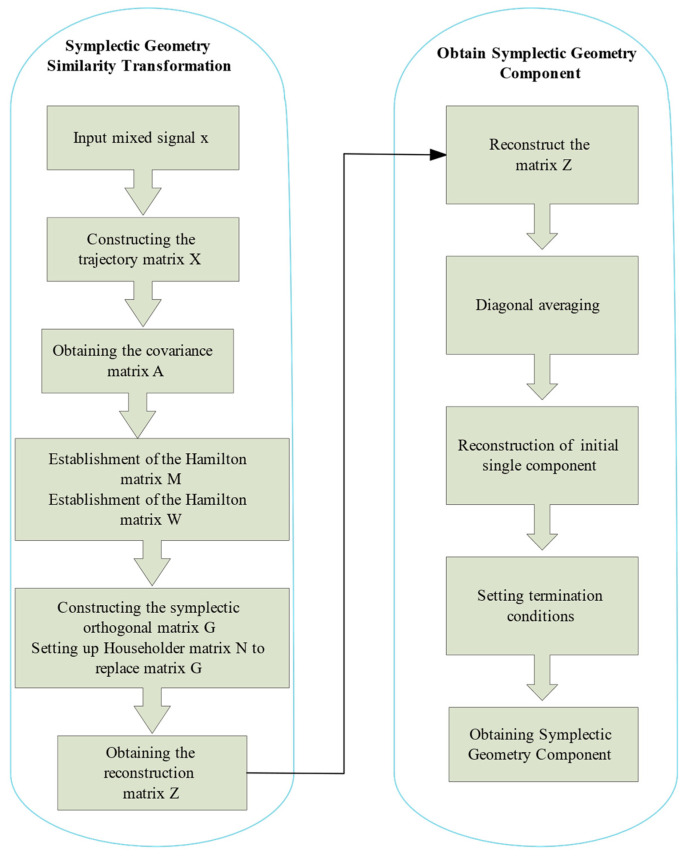
SGMD decomposition mixed signal principle.

**Figure 2 sensors-24-00462-f002:**
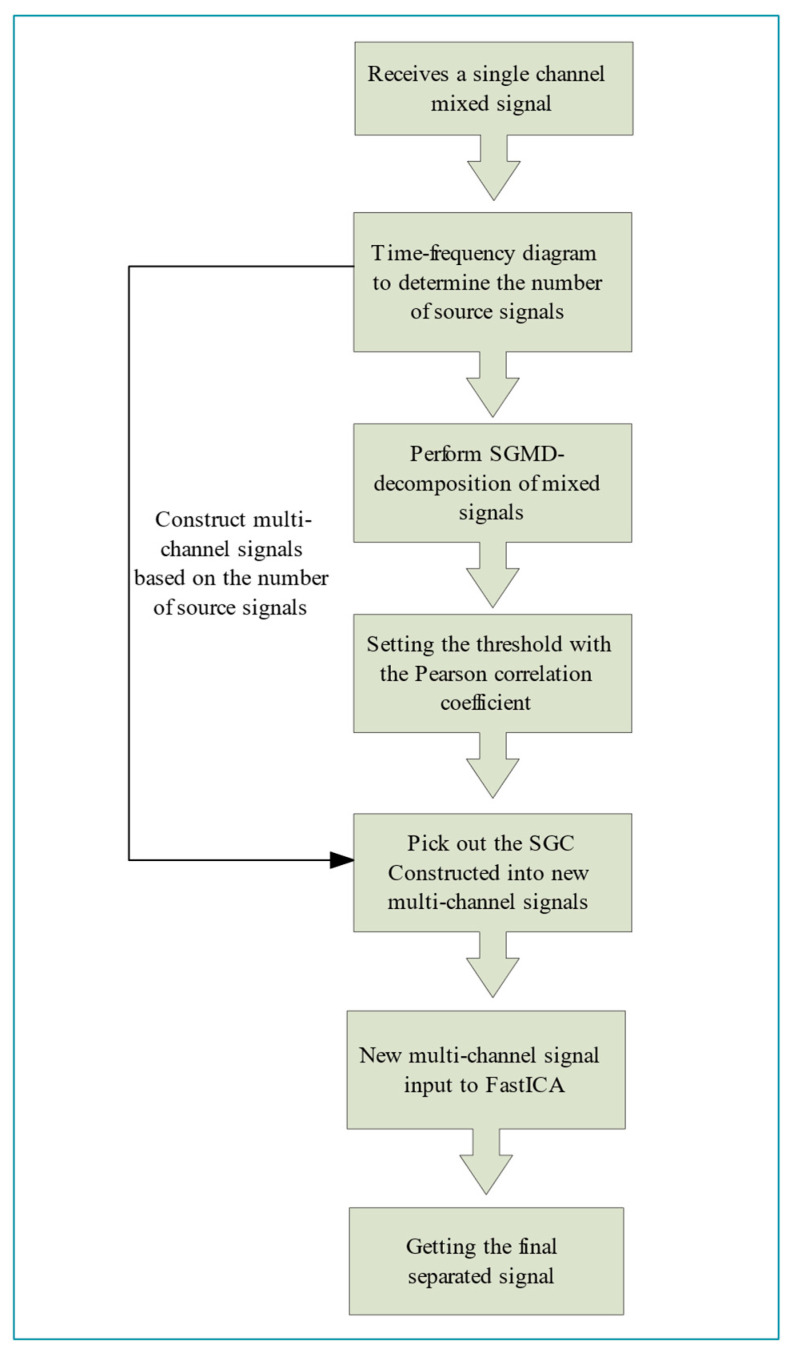
Steps of SGMD-FastICA comprehensive separation algorithm.

**Figure 3 sensors-24-00462-f003:**
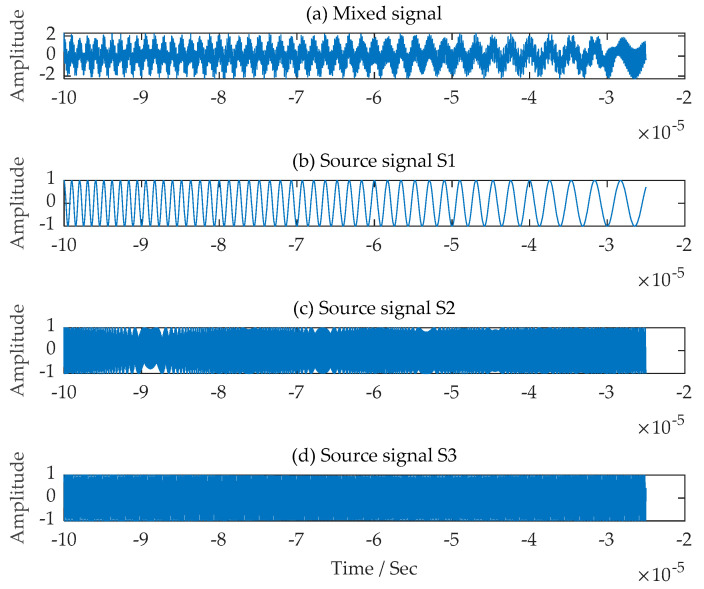
Time-domain waveforms of a single-channel mixed signal and three source signals. (**a**) Single-channel mixed signal; (**b**) The first LFM source signal; (**c**) The second LFM source signal; (**d**) The FM source signal.

**Figure 4 sensors-24-00462-f004:**
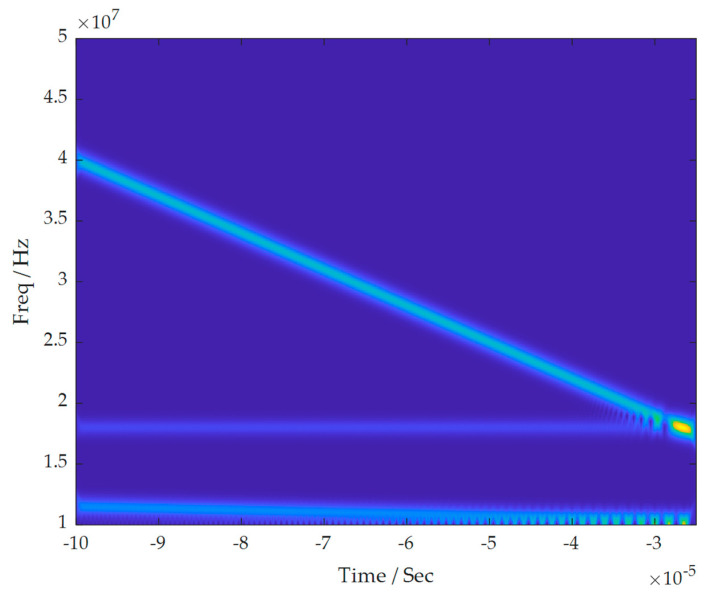
Time-frequency diagram of single-channel mixed signal.

**Figure 5 sensors-24-00462-f005:**
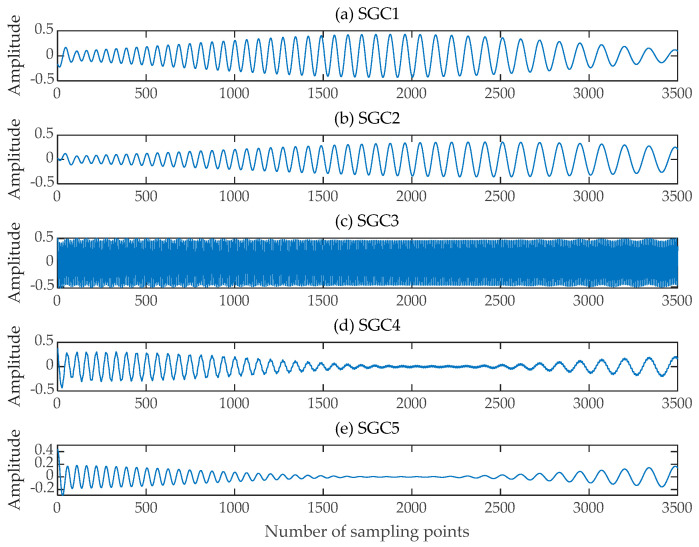
Symplectic Geometry Components. (**a**–**e**) Five SGCs with Pearson correlation coefficients greater than 0.45.

**Figure 6 sensors-24-00462-f006:**
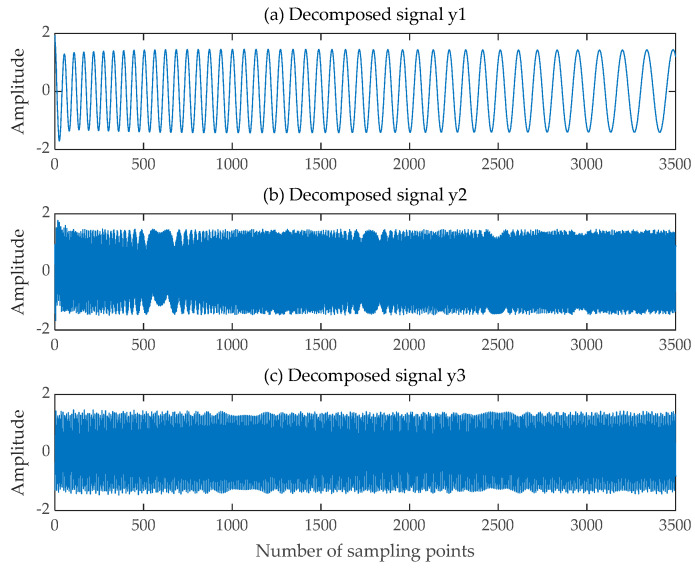
Decomposed signals obtained by SGMD-FastICA comprehensive separation algorithm. (**a**) The first decomposed signal; (**b**) The second decomposed signal; (**c**) The third decomposed signal.

**Figure 7 sensors-24-00462-f007:**
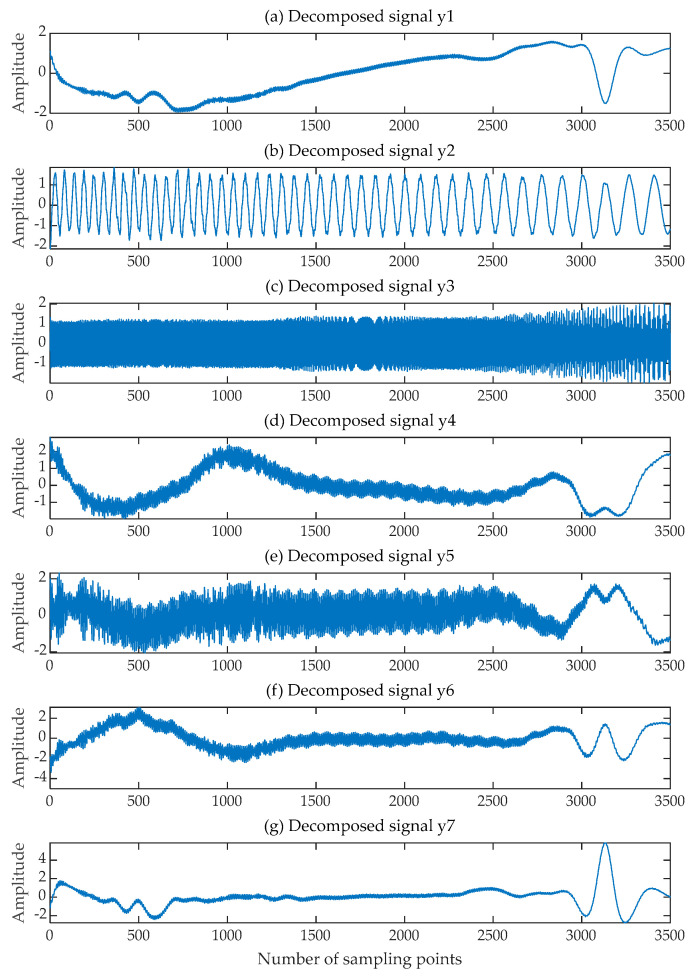
Decomposed signals obtained by EMD-FastICA. (**a**–**g**) Seven decomposed signals.

**Figure 8 sensors-24-00462-f008:**
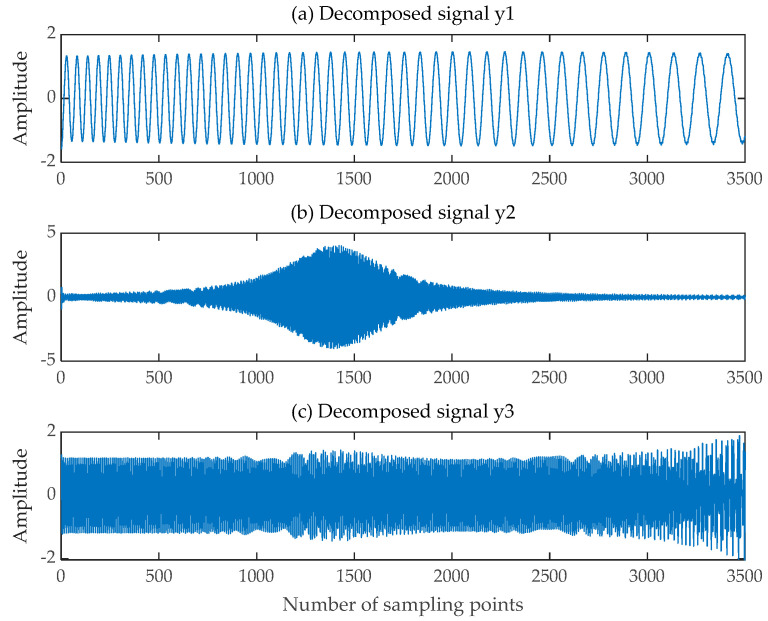
Decomposed signals obtained by VMD-FastICA. (**a**) The first decomposed signal; (**b**) The second decomposed signal; (**c**) The third decomposed signal.

**Figure 9 sensors-24-00462-f009:**
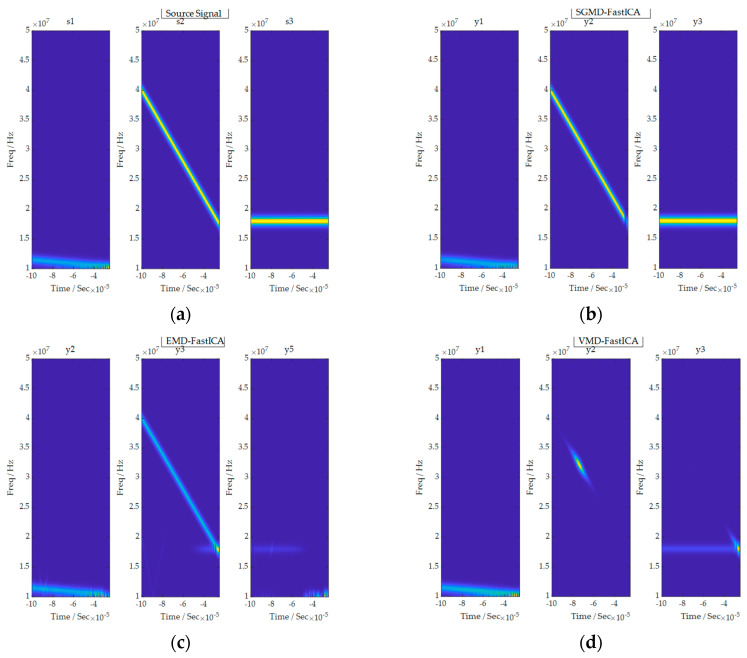
Time-frequency diagram corresponding to the source signal and the decomposed signals obtained by the three comprehensive separation algorithms. (**a**) Time-frequency diagram of the source signal; (**b**) Time-frequency diagram corresponding to the decomposed signal obtained by SGMD-FastICA; (**c**) Time-frequency diagram corresponding to the decomposed signal obtained by EMD-FastICA; (**d**) Time-frequency diagram corresponding to the decomposed signal obtained by VMD-FastICA.

**Figure 10 sensors-24-00462-f010:**
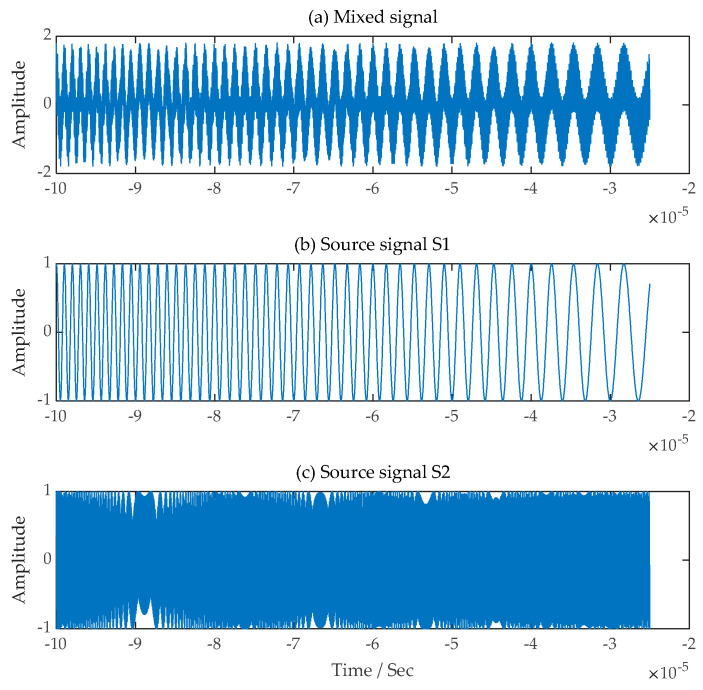
Time-domain waveforms of a single-channel mixed signal and two source signals. (**a**) Single-channel mixed signal; (**b**) The first LFM source signal; (**c**) The second LFM source signal.

**Figure 11 sensors-24-00462-f011:**
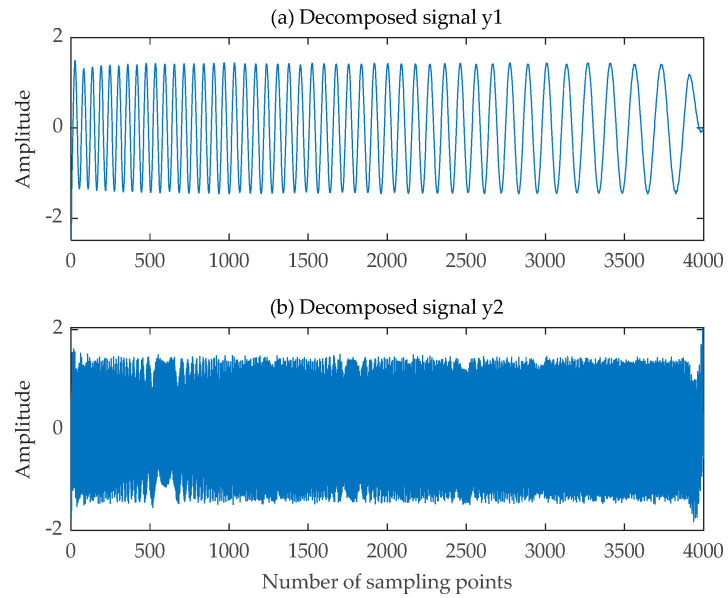
Decomposed signals obtained by SGMD-FastICA. (**a**) The first signal obtained in a Gaussian white noise environment; (**b**) The second signal obtained in a Gaussian white noise environment.

**Figure 12 sensors-24-00462-f012:**
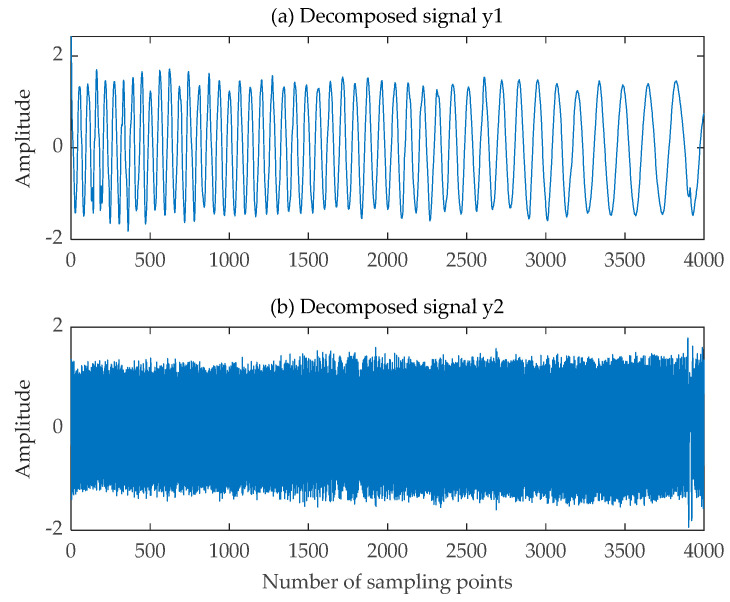
Decomposed signals obtained by EMD-FastICA. (**a**) The first signal obtained in a Gaussian white noise environment; (**b**) The second signal obtained in a Gaussian white noise environment.

**Figure 13 sensors-24-00462-f013:**
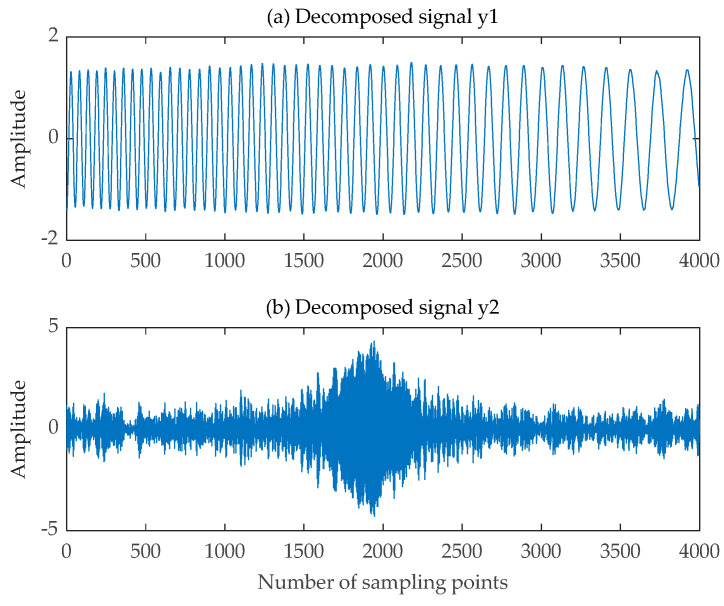
Decomposed signals obtained by VMD-FastICA. (**a**) The first signal obtained in a Gaussian white noise environment; (**b**) The second signal obtained in a Gaussian white noise environment.

**Figure 14 sensors-24-00462-f014:**
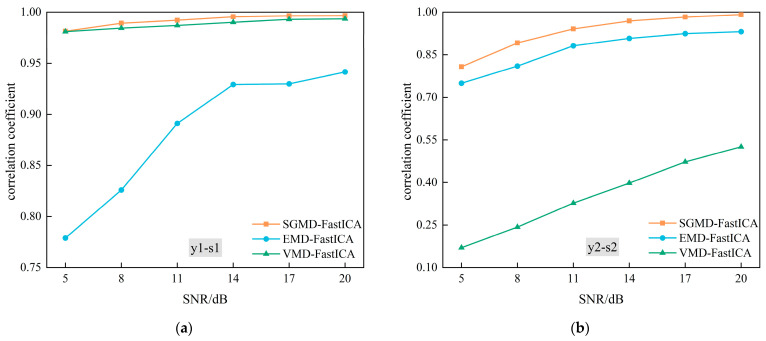
Correlation coefficients of the three comprehensive separation algorithms. (**a**) Correlation coefficients between the first source signal and its corresponding first separated signal obtained using the three algorithms in turn; (**b**) Correlation coefficients between the second source signal and second separated signal obtained using the three algorithms in turn.

**Figure 15 sensors-24-00462-f015:**
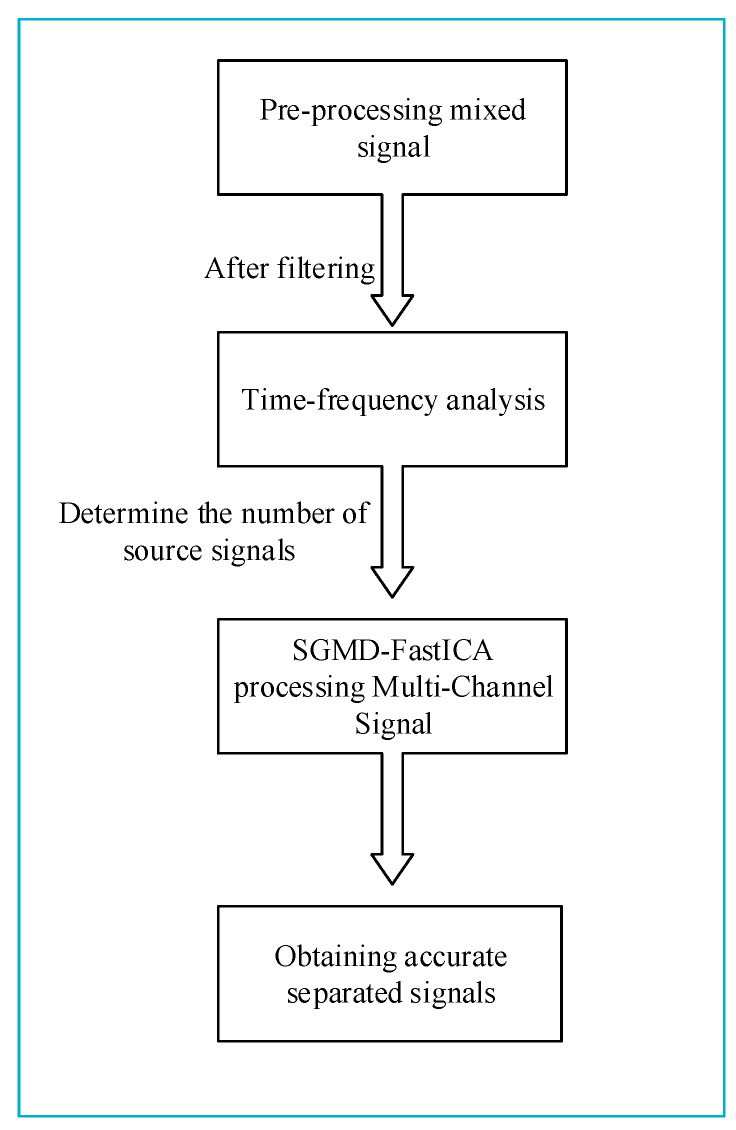
Improved SGMD-FastICA comprehensive separation algorithm.

**Figure 16 sensors-24-00462-f016:**
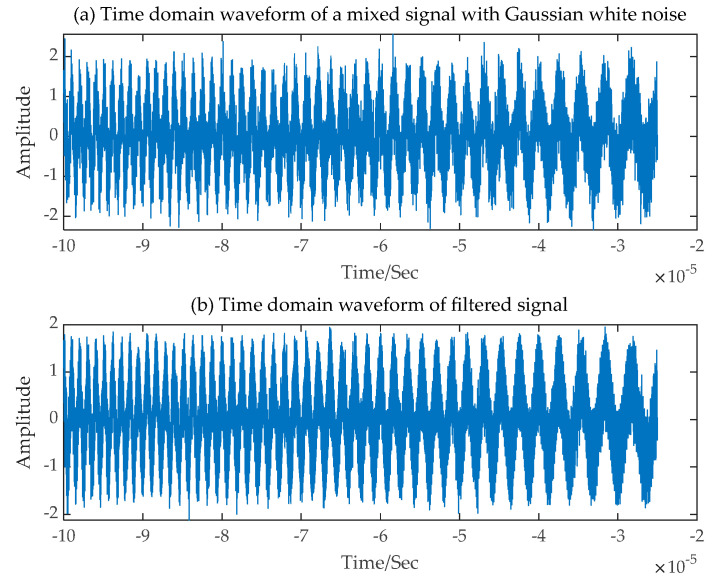
Noise-containing mixed signal and pre-processed mixed signal at 0 dB SNR. (**a**) Noise-containing mixed signal; (**b**) Pre-processed mixed signal.

**Figure 17 sensors-24-00462-f017:**
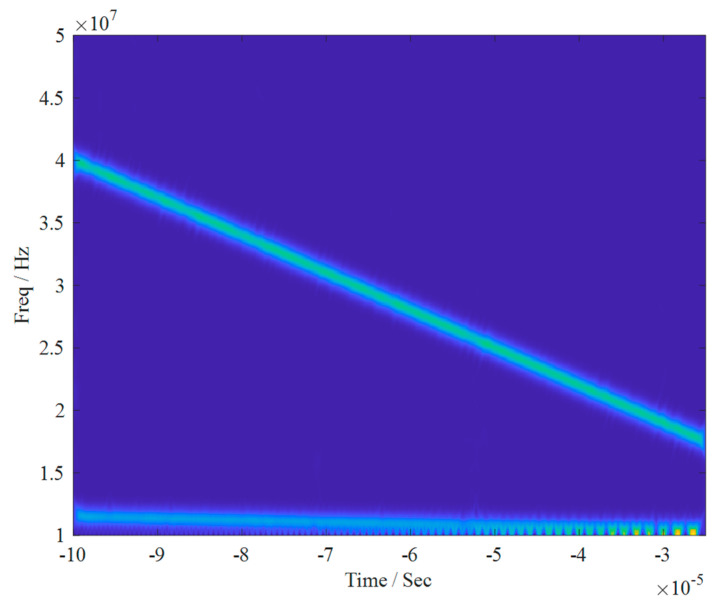
Time-frequency diagram of denoised mixed signals.

**Figure 18 sensors-24-00462-f018:**
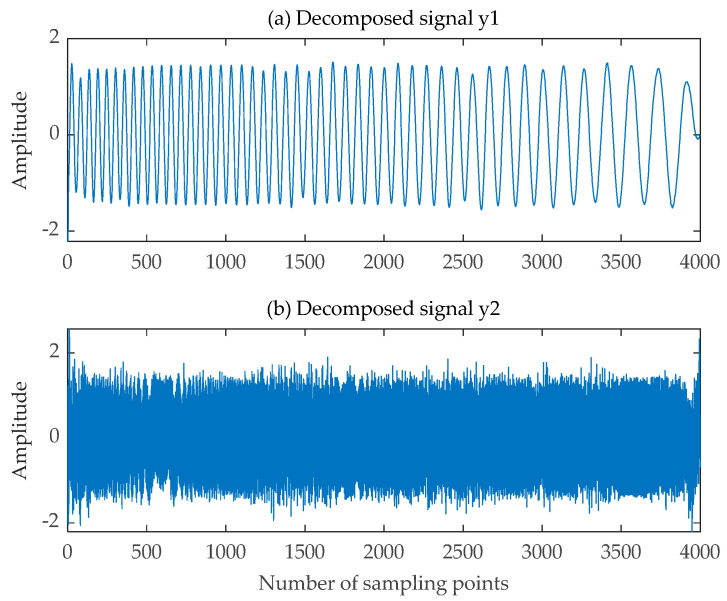
Decomposed signal obtained by the improved SGMD-FastICA comprehensive separation algorithm. (**a**) The first decomposed signal; (**b**) The second decomposed signal.

**Figure 19 sensors-24-00462-f019:**
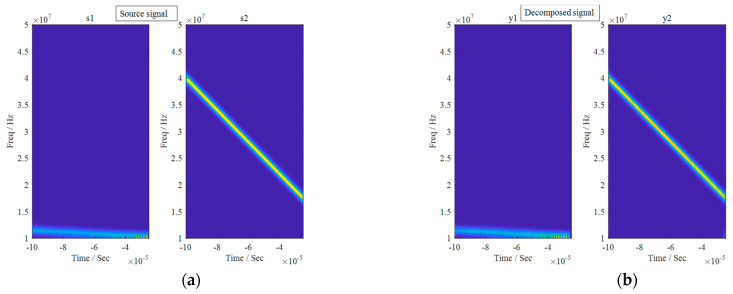
Time-frequency diagram of the source and decomposed signals. (**a**) Time-frequency diagram corresponding to the source signal; (**b**) Time-frequency diagram corresponding to the decomposed signal.

**Figure 20 sensors-24-00462-f020:**
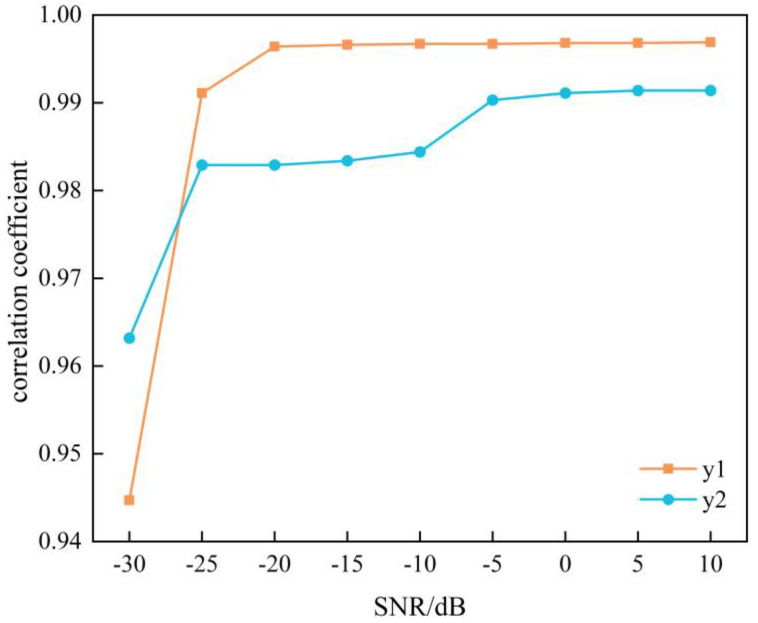
Correlation coefficients of two LFM signals under different SNR conditions.

**Table 1 sensors-24-00462-t001:** Matrix of correlation coefficients for the three comprehensive separation algorithms.

Total Correlation Coefficient Matrix			
SGMD-FastICA	S1	S2	S3
y1	0.99790	0.00300	0.00020
y2	0.00001	0.99190	0.03590
y3	0.00040	0.01210	0.97120
EMD-FastICA	S1	S2	S3
y2	0.91570	0.00830	0.05410
y3	0.01050	0.95160	0.14850
y5	0.21180	0.01600	0.57450
VMD-FastICA	S1	S2	S3
y1	0.99740	0.00870	0.00250
y2	0.00005	0.63270	0.05260
y3	0.00010	0.20900	0.86400

**Table 2 sensors-24-00462-t002:** Correlation coefficients of the three separation algorithms in a 20 dB SNR environment.

Correlation Coefficient	SGMD-FastICA	EMD-FastICA	VMD-FastICA
Coefficient I	0.9964	0.9415	0.9900
Coefficient II	0.9910	0.9313	0.5257

**Table 3 sensors-24-00462-t003:** Matrices of correlation coefficients obtained from the improved SGMD-FastICA comprehensive separation algorithm at 0 dB signal-to-noise ratio.

Correlation Coefficient	S1	S2
**y1**	0.9967	0.0034
**y2**	0.0004	0.9895

## Data Availability

Data are contained within the article.

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
