# Peer review of "Comprehensive Separation Algorithm for Single-Channel Signals Based on Symplectic Geometry Mode Decomposition"

_sensors, 2024, doi:10.3390/s24020462_

Round 1

Reviewer 1 Report

Comments and Suggestions for Authors

This paper presents a comprehensive single-channel mixed signal separation algorithm based on the combination of Symplectic Geometry Mode Decomposition (SGMD) and FastICA algorithm. There are certain queries that needs to be addressed:

1. The test signals are mostly quite simple to separate and there is no challenging case like crossing chirps and one or multiple points and sinusoidal frequency modulated signals, in time-frequency domain.

2. There is no comparison or discussion on some of the modern time frequency methods used for similar tasks like proposed by Amin, Boashash, Cohen, Stankovic, Shafi & Irena and others. A search on these methods can be useful and related discussion must be added.

3. The novelty of the proposed method while considering combination of the two existing algorithms is also questionable and shall be elaborated for the interested reader.

4. I would also be interested to see some complexity analysis of the proposed approach vis a vis other existing methods.

5. The reference section needs to be beefed up with the latest references along with related discussion within Introduction and Literature review sections.

6. The methodology section is not well written and does not convey the details of the adapted approach.

7. The claim that the SGMD algorithm could eliminate noise interference while keeping the raw time series unchanged, achievable through symplectic geometry similarity transformation during the decomposition of mixed signals, is questionable as the SNR is not well described.

Comments on the Quality of English Language

Minor English editing is required.

Reviewer 2 Report

Comments and Suggestions for Authors

The paper presents a method for separation of signals from a single channel source based on symplectic geometry mode decomposition and the fast-ICA algorithm.  The algorithm is demonstrated to work well on some cases better then pre-existing algorithms from the literature.  I have a few concerns that I feel should be addressed before final publication of the paper:

1. Single channel source separation algorithms/system must make some assumption about the structure of the different signals that are to be separated.  What are the assumptions made by the algorithm in this paper? What are the types of signals that the proposed algorithm would not be able to separate versus the ones that it can separate? How do these assumptions correspond to those of the algorithms that are being compared with this one?

2. The algorithm is shown being applied to LFM signals. It seems that these signals are being sampled before being processed but there is no information on the sampling rate, record length, and frequency values of the signal.  These should be provided. Some values are provided but not enough for the reader to precisely generate the same signals.

3. The paper should supply all parameters of the algorithm used to perform the decompositions shown in Figure 3 to Figure 20.  These should be provided so the interested reader can reproduce the presented results. For example, the SGMD algorithm shown in Section 2.2 has several parameters such as tau, , n, d^*, and m^*.  What are these values set to during the processing?

4. Figure 2 has a block labelled with "Time-frequency diagram to determine number of source signals".  What is the analysis done here to determine the number of signals?  At minimum, a reference to a paper describing this should be given.

This paper presents some interesting results that I believe will be of interest to many but more information is needed so a reader can determine if this method could apply for their application or check if they implemented the algorithm correctly.

Comments on the Quality of English Language

I only found minor language grammar issues at one or two points.  Another proofread would be a good idea but the vast majority of the paper has more than acceptable English usage.

Round 2

Reviewer 1 Report

Comments and Suggestions for Authors

Overall, I see major improvements, however, not all TF methods mentioned in the first round of review are addressed.

Comments on the Quality of English Language

Minor English issues

Reviewer 2 Report

Comments and Suggestions for Authors

The authors have answered most of my questions about their method. However, I think that they still need to give more information on the assumptions behind the method.  The stated assumptions of each of the sources being statistically independent and being of non-Gaussian distribution are far too general to answer the question of the assumptions.  The problem is that these are the standard assumptions of ICA.  The authors say that the problem cannot separate signals which are 'too similar' but does not state what this means in terms of statistics or distributions.  The authors should expand on this.  I think that the authors have a good method which will be useful to many but the exact nature of when the method is useful needs to be defined.

Comments on the Quality of English Language

The language is of good quality.
